# Implementing a Personalized Antimicrobial Stewardship Program for Women with Gynecological Cancers and Healthcare-Associated Infections

**DOI:** 10.3390/jpm12040650

**Published:** 2022-04-18

**Authors:** Simona Di Giambenedetto, Alberto Borghetti, Lorena Quagliozzi, Valeria Gallucci, Francesca Lombardi, Arturo Ciccullo, Anna Fagotti, Enrica Tamburrini, Giovanni Scambia

**Affiliations:** 1UOC Malattie Infettive, Dipartimento di Scienze di Laboratorio ed Infettivologiche, Fondazione Policlinico Universitario A, Gemelli IRCCS, 00168 Roma, Italy; simona.digiambenedetto@unicatt.it (S.D.G.); enrica.tamburrini@policlinicogemelli.it (E.T.); 2Dipartimento di Sicurezza e Bioetica Sezione Malattie Infettive, Università Cattolica del Sacro Cuore, 00168 Roma, Italy; francesca.lombardi@policlinicogemelli.it; 3UOC Ginecologia Oncologica, Dipartimento di Scienze della Salute della Donna, del Bambino e di Sanità Pubblica, Fondazione Policlinico Universitario A, Gemelli IRCCS, 00168 Roma, Italy; lorena.quagliozzi@policlinicogemelli.it (L.Q.); valeria.gallucci@policlinicogemelli.it (V.G.); anna.fagotti@policlinicogemelli.it (A.F.); giovanni.scambia@policlinicogemelli.it (G.S.); 4UOC Malattie Infettive, Ospedale San Salvatore, 67100 L’Aquila, Italy; arturo.ciccullo@gmail.com

**Keywords:** gynecologic oncology, personalized medicine, healthcare-associated infections, antimicrobial stewardship, OPAT

## Abstract

Healthcare-associated infections (HCAIs) represent a major cause of morbidity and mortality in gynecologic cancer patients, requiring personalized cures. A retrospective study on gynecologic patients with HCAIs, managed through an antimicrobial stewardship program, was performed, focusing on rates of clinical cure, breakthrough/relapse of infections, death, and time of hospital stay (THS). In total, 27 patients (median 60 years, mainly suffering from ovarian, cervical, and uterine cancer) were evaluated by a specialist in infectious diseases and were mainly diagnosed with complicated urinary tract (cUTIs, 12 cases, 44.4%) and bloodstream infections (BSIs, 9 cases, 33.3%). A total of 15 cases (11 cUTIs, 73.3%) were managed with no need for hospitalization and received a median of 11 days of outpatient parenteral antimicrobial therapy (OPAT). In the remaining 12 cases (BSIs in 8 cases, 66.7%), the median THS was 11 days, with 15 days median overall duration of antimicrobial therapy (median 5-day reduction in THS). The management of patients also included source control and wound care. All patients reached clinical cure, with no case of breakthrough infection, one case of relapse, and one death within 30 days (not attributable to the infection). HCAIs in patients with gynecologic tumors can be managed through a patient-centered, multidisciplinary antimicrobial stewardship program.

## 1. Introduction

Gynecologic cancer patients have specific needs to be met, in order to benefit from the best therapeutic outcomes during the whole care process [1]. Besides target therapies, psychological and spiritual support, gynecological, genetic, and nutrition counselling, patients could experience fragmentation of services if the care center does not provide integrated management. Additionally, phenotype patterns may differ among patients, hence adequate monitoring programs should be implemented.

The centralization of high-volume centers is fundamental to offering integrated diagnostic and personalized therapeutic pathways that optimize clinical resources while improving patients’ quality of life. Among patients’ basics needs, the management of healthcare-associated infections (HCAIs) must be addressed since infectious diseases have been recognized as a major cause of morbidity and mortality in gynecologic cancer patients [2,3], with several factors contributing to the explanation of this phenomenon, such as the immunosuppression associated with both the disease and the chemotherapy, the presence of indwelling urinary and vascular catheters or specific surgical procedures [2]. 

A recent study [1] showed that an Enhanced Recovery After Surgery (ERAS) program has been implemented in our research hospital (Fondazione Policlinico Universitario A. Gemelli IRCCS, Roma, Italy), leading to a reduction in the length of hospital stay, perioperative complications, and costs [4]. In relation to infectious diseases, this program had the potential to reduce the risk of post-operative lung, urinary tract, surgical site infections, etc. [5]. However, despite advances in surgical techniques, antimicrobial prophylaxis, and the EARS program, another study showed a 6.1% incidence of healthcare-associated infections in women with gynecologic malignancies between 2016 and 2017 [6]: among 68 women admitted to the emergency department (ED) with a clinical suspicion of bloodstream infection (BSI), 34% had confirmed positive blood culture and 31% a urinary tract infection (UTI). Moreover, infections represented the major cause of unplanned readmission after gynecologic surgery in 45% of patients in a multicohort analysis by Uppal et al. [7].

The issue of HCAIs can be difficult to face due to the increasing spread of antimicrobial-resistant organisms in the European Union countries. In this scenario, Italy has the highest burden of deaths and disability-adjusted life years (DALYs) due to antimicrobial-resistant bacteria among all European countries [8]. This could, therefore, minimize the efforts of a successful program to manage gynecologic cancer patients by increasing the rate of readmissions, the length of hospital stay, and their psychological condition.

To address the problem of antimicrobial resistance and HCAIs, centers for disease control and prevention recommend the implementation of antimicrobial stewardship programs that consist of systematic measurement and coordinated interventions designed to promote the optimal use of antimicrobial agents (including their choice, dosing, route, and duration of administration), to optimize clinical outcomes while minimizing unintended side effects of antimicrobial use [9]. As requested and verified by the Joint Commission International accreditation recently obtained by our hospital, an antimicrobial stewardship program had already been implemented. However, specific interventions for gynecological cancer patients were missing. The possibility of implementing such a program to provide early recognition and treatment of infectious complications could, therefore, be of particular value for this vulnerable population, as patients in this group need integrated and participative approaches for managing complications that can be detrimental to their health from a physical and psychological perspective. 

Patients with comorbidities (such as gynecological cancer and infections) require further dedicated approaches since their care encompasses multiple levels and conditions along with mixed solutions for their complex situation. The intricate challenge of clinical complexity requires to be appraised, understood, and formulated with attention to the severity of the situation in each patient’s life. A major form of complexity was brought about by COVID-19 pandemics because clinical management and care become a major global challenge for personalized approaches that help patients achieve resilience and a better quality of life [1].

A recently implemented set of interventions was based on the presence of an infectious disease specialist specifically in charge of the gynecologic wards, who performed a bedside evaluation of each patient while making shared decisions with caring gynecologists and patients about the need for hospitalization, the choice personalized antibiotic therapies, the switch from intravenous to oral therapy, and hospital discharge. Different strategies (described in Section 2.2. Interventions) were put in place with the support of trained nurses who followed up with the patients outside the hospital, and clinical pharmacists that allowed using elastomeric pumps for all-day infusions of different antibiotics (usually not available for outside hospital usage). Together with the dedicated patient and caregivers’ education on self-administer antibiotics, these strategies adhere to the principles of “P4 medicine” (predictive, participative, preventive, and personalized) while leading to the active involvement of the caring physicians, specialists in infectious diseases, nurse staff, and patients and their families [1].

We hereby present data on the clinical outcomes of this systematic and multidisciplinary approach to gynecologic cancer patients with a suspected or confirmed infection and who were followed up for at least one month.

## 2. Materials and Methods

### 2.1. Study Population

Patients who were followed up at our center for the treatment of the oncologic disease and diagnosed with an infection in different clinical contexts (gynecologic ambulatory/day hospital, gynecologic ward, emergency department) and requiring antimicrobial therapy were eligible for study analysis. The study period lasted from 1 September to 30 November 2021. Exclusion criteria were as follows: less than 30 days follow-up or lack of information about clinical status at 30 days since the diagnosis of infection. 

### 2.2. Interventions

At our hospital, the evaluation of patients with a suspect or diagnosis of infection is performed by a specialist in infectious diseases who is specifically in charge of evaluating gynecologic patients (with or without cancers) in different clinical contexts, from the ambulatory/day hospital to the different gynecologic wards, and who is available 7 days a week during daily shift (for night emergencies a specialist in infectious diseases is always available on request).

The specialist role is to direct the diagnostic and therapeutic processes when a patient is suspected to have an infection, by facilitating interprofessional communication within the hospital. Initially, specialists assess whether a patient needs hospitalization for the suspected infection or not. Then, they indicate a set of diagnostic tests according to the situation of each patient and the clinical context: the diagnosis of infection is performed through a patient-centered approach consisting of clinical history, physical examination, and specific microbiological tests or imaging (e.g., blood cultures, urine cultures, chest X-ray or chest/abdominal CT scan), in order to maximize the beneficial outcomes. 

In our center, a “fast microbiology” service is also in place. Particularly, after a blood culture demonstrates bacterial growth, together with a classic culture method with susceptibility test performance, a direct microorganism identification method, i.e., MALDI-TOF (MALDI Biotyper^®^—Bruker Daltonics Inc., Billerica, MA, USA), an automated next-generation microbial identification system based on matrix-assisted laser desorption-ionization time-of-flight, allows for rapid identification of the bacterial species. Additional tests, such as the CTX-M MULTI immunochromatographic (NG Biotech, Guipry, France), the immunochromatographic NG-Test Carba 5 (NG Biotech, Guipry, France), and the Eazyplex MRSA (AmplexDiagnostics GmbH, Gars-Bahnhof, Germany) are also available to detect the most frequent resistance genes encountered in our specific epidemiological setting, whereas a multiplex PCR system (FILMARRAY™ Blood Culture Identification Panel, bioMérieux—Pioneering Diagnostics, Salt Lake City, UT, USA), allows for either species and resistance genes recognition in case of polymicrobial bacterial growth. After recognition of the specific pathogen from blood culture, clinical microbiologists immediately transmit the information to the infectious disease specialist daily, as established by the internal Antimicrobial Stewardship Program. 

In this way, the specialist is constantly informed about positive blood cultures, allowing for early detection, communication, and targeted antimicrobial therapy of severe infections. Furthermore, the specialist actively re-evaluates the need for antimicrobial therapy on a daily basis and chooses when to de-escalate therapy from intravenous to an oral way of administration.

For patients who are not candidates for hospitalization or prolongation of hospitalization but need intravenous antibiotic administration, a hotel infrastructure within the hospital campus (specifically available for gynecological patients living outside the region and requiring short term re-assessment [1]) is used to provide outpatient antimicrobial parenteral therapy (OPAT), and wound care for surgical site infections by trained nurses, aiming at reducing expenditures and care fragmentation. Whenever possible and necessary, a caregiver is specifically trained on how to administer antibiotics to the patient on certain days of the week through the indwelling vascular catheter, to limit the need for ambulatory access and everyday nurse visit. 

For some antibiotics, usually requiring multiple in-hospital daily administrations (such as piperacillin/tazobactam or cefepime), elastomeric pumps for all-day-long infusions were provided upon request, after approval from the hospital pharmacists, which confirmed the compound stability during the whole time of infusion.

#### 2.2.1. Outcomes

The main outcomes of the study were clinical cure of the infectious disease at the end of antimicrobial therapy (e.g., lack of signs/symptoms consistent with the infection), infection breakthrough or relapse within 30 days, and death within 30 days. Additionally, we considered the length of hospital stay attributable to the infectious disease for patients already hospitalized for other reasons or requiring hospitalization because of infection.

#### 2.2.2. Statistical Analysis

This was a monocentric retrospective observational study. A descriptive analysis of the study population was performed, and a description of the outcomes was determined. Given the limited sample size of the study population and the low incidence of adverse outcomes, no specific association could be investigated between patients’ characteristics, type of infection, and the study outcomes.

#### 2.2.3. Ethics

The study was performed in accordance with the 1964 Declaration of Helsinki and later amendments. All patients signed an informed consent form for use of their clinical and laboratory data using an aggregated and anonymous form. Access to the database and data analyses are regulated by Fondazione Policlinico Gemelli Ethics Committee, that approved the study design (protocol identification number: 4582) on the 16 November 2021, and by Italian and European privacy legislation.

## 3. Results

A total of 27 patients (median age 60 years, IQR 49–70) were evaluated by a specialist in infectious diseases in the following clinical settings: 10 in gynecologic wards, 8 in gynecologic ambulatory/day hospital setting, and 9 at the ED. In total, 31 patients were excluded since the reason for infectious disease consultation did not concern an HCAI (26 patients, 83.9%) or information was not available since follow-up (5 patients, 16.1%).

Table 1 summarizes the main clinical and demographical characteristics of the study population. 

In 15 cases, there was no need for hospitalization since the patient could be managed through OPAT; in this setting, the infections were mainly represented by UTIs (11 cases, 73.3%), followed by BSI, PID, pneumonia, and rectovesical fistula (1 case each, 6.7%). The median duration of antimicrobial therapy in this group was 11 days (6–17).

In the remaining 12 cases, the infections required hospitalization or prolongations of the current hospital stay. These infections were represented by BSI in eight cases (66.7%), UTI and PID in one case each (8.3%), SSIs in two cases (16.7%). Antimicrobial therapy was administered for a median of 15 days (IQR 10–17), 9 of which (IQR 1–11) fell during the hospital stay. This led to a significant reduction in the length of hospital stay (5-day median reduction, IQR 4–9).

No patient was managed with oral antimicrobial therapy, due to the high rate of resistant bacterial species (specifically, 18 infections were caused by Enterobacterales species, 61.1% of which were ceftriaxone- and/or carbapenems-resistant). The most used antibiotics in OPAT were carbapenems (ertapenem in 15 cases, 55.6%, and thrice-a-day meropenem in 1 case, 3.7%), followed by other beta-lactams (ceftriaxone in 4 cases, 14.8%, piperacillin/tazobactam in continuous infusion through an elastomeric pump in 3 cases, 11.1%), glico-lipopeptides (daptomycin and teicoplanin in 1 case each, 3.7%), amikacin and colistin (1 case each, 3.7%).

Apart from antimicrobial therapy, the management of patients also included source control, which consisted of removal of the indwelling vascular catheters when these were considered the source of infection (five cases, 18.5%), and removal of urinary catheters when they were associated with UTIs and wound care. In the hospital setting, these interventions were performed by the ward nursing staff, whereas for non-hospitalized patients, this was performed by two nurses specifically trained and assigned to OPAT. Outpatient management was performed in the dedicated facility near the hospital with periodic nurse visits for antimicrobial therapy, wound care, and blood sampling.

Overall, the outcomes were excellent, with 27 cases (100%) of clinical cure (e.g., lack of signs/symptoms consistent with the infection) at the end of treatment, no case of breakthrough infection, 1 case (3.7%) of relapse and 1 case of death within 30 days. Notably, the single case of death was attributed to the terminal phase of the oncologic disease, not the infection.

## 4. Discussion

Gynecologic cancers are the most common female cancer and a major source of mortality and morbidity [10,11,12,13,14,15,16,17]. Clinical management requires tailored solutions since neoplastic diseases challenge their quality of life, leading to biographical breakdowns. Comorbid conditions such as infections may interfere with oncological management, limit the availability of hospital beds, and require more economic resources for antimicrobial therapy; in addition, it is detrimental to patients’ physical and psychological health. This complexity should be assessed, understood, and treated in accordance with several aspects and levels, to tailor the clinical tools for effective care management. Some cancer patients may require multimodal therapies that interact in complex and different ways for each case; therefore, a dedicated health professional who knows the specific case of each patient may help achieve the main aim of personalized medicine—namely, providing “the right treatment for the right patient, at the right time”. As a result, conventional healthcare paradigms focusing on the disease and immediate standard care are often seen as inadequate.

Antimicrobial stewardship programs based on infectious disease consultation have been demonstrated to be effective in optimizing antibiotic consumption, reducing the incidence of *Clostridium difficile* infection and antimicrobial resistance patterns, without affecting patient outcomes [10,11]. Clinical studies have increased in recent years that concern antimicrobial stewardship programs specifically addressing oncologic populations, which confirm the potential usefulness of several approaches in these programs to decrease antibiotic use without increased negative consequences. However, more quality data are needed, especially in the management of more specific cancers, such as gynecologic cancers [12].

Apart from infectious disease consultation, one of the most important factors affecting patients’ outcomes is represented by delivering a multidisciplinary approach to the management of infections. A previous report by Thursky et al. [13] provided evidence of the usefulness of implementing an integrated approach for the management of sepsis in a cancer hospital: After the implementation period, the rate of time to antibiotics was halved, and the rate of intensive care unit admission was reduced, as well as the overall length of stay. Although a control arm was not present in our study, the possibility of reducing the time of hospital-based antimicrobial administration seems to suggest a similar outcome in gynecologic cancer patients.

Integrating a structural model for each gynecological cancer patient with infections is essential to meet their specific needs, thus providing personalized care. To fulfill this objective, strict adherence to infectious disease specialist indications is required [14], and appropriate tools are necessary. For example, the use of outpatient parenteral antimicrobial therapy, defined as the administration of parenteral antimicrobial therapy in at least two doses on different days without intervening hospitalization [15], is a potential tool to reach this goal [16], especially in a context of a high incidence of antimicrobial-resistant bacterial species. In fact, despite the non-negligible proportion of gynecological cancer patients with severe infectious complications, a new hospitalization or the prolongation of the current one could not be the more suitable option in every case.

As demonstrated in previous studies [17,18], OPAT is feasible for the treatment of different infectious diseases, it can be cost-effective, compared with inpatient care, and improves patient satisfaction. In our preliminary experience, we performed a diagnostic and treatment stewardship program grounded on the idea of a dedicated healthcare professional (the infectious disease specialist for gynecological cancer patients) as a pivotal figure in managing the whole care process, resulting in a high rate of definite diagnosis and targeted therapies, with the outcome of providing the best cure for patients while minimizing the need for prolonged hospitalization. For background, it should be noted that this program is integrated into a broader organizational personalized medicine infrastructure dedicated to gynecologic oncology, which recently redefined its care planning for the pandemic to reduce the in-hospital spread of the virus [1].

Overall, the observations of this study highlight the possibility of treating mild/moderate to serious infections in patients with gynecological tumors through a patient-centered, multidisciplinary approach. Most infections (especially UTIs) were managed with no need for hospitalization, whereas more serious conditions were addressed with a few days of hospitalization or prolonged stay, allowing earlier discharge once patients were stabilized. The high percentage of clinical cure and lack of infection relapse reported in our study seems to be in line with other experiences: Despite differences in antimicrobial stewardship approaches, the rate of mortality or unfavorable outcomes (for instance, ICU admission, re-escalation of antibiotics, etc.) was not higher in stewardship programs, compared with control groups, according to a recent review of the literature [13].

Importantly, despite potential difficulties in OPAT administration (such as administering multiple daily doses of antibiotics, the impact of real-life temperature variations, a sub-optimal clinical and laboratory follow-up [16], etc.), outcomes of our experience demonstrate most of these issues can be overcome, and a high percentage of clinical cure with very low rates of disease relapse/breakthrough can be reached. These results seem to be encouraging, considering the high rate of infections by multi-drug-resistant bacteria that were treated. Considering that the challenge of antimicrobial resistance in Italy leads to dramatic consequences [8], administering antibiotics for hospital use in outpatient facilities may have the double benefit of providing the best therapeutic option while reducing the risk of becoming colonized with multi-drug-resistant bacteria.

To our knowledge, this is the first study exploring the effect of an antimicrobial stewardship program dedicated to gynecological cancer patients. In particular, we highlighted the possibility of implementing more stewardship interventions at the same time, including personalized diagnostic approaches and antibiotic therapy, early antibiotic de-escalation, use of OPAT for reducing hospital stay, and patient evaluation by an infectious disease specialist after hospital discharge.

Some limitations merit discussion. First, the small sample size of our population hampers definite conclusions about the effectiveness of this strategy, especially in comparison with previous approaches. More importantly, given the retrospective nature of the study, there was a lack of patients’ reported outcome measures (PROMs), which could represent one of the most important treatment outcomes, considering the particular clinical setting and the specific necessities of vulnerable patients [1]. Moreover, a pharmacoeconomic assessment could help understand the cost-effectiveness of this program: In fact, costs associated with this personalized approach could become excessive when considering a population larger than that included in our study; the availability of dedicated health-care professionals or larger infrastructures could become other important limits for implementing this care model. Despite these considerations, our data encourage longer follow-up studies to better understand the management of patients who experience healthcare-associated infections.

Future studies will include PROMs and patient-reported experience measures (PREMs), in order to achieve a holistic vision of how this program may impact patients’ experiences with these types of illnesses. In the field of precision oncology, translational research could ameliorate clinical management of gynecological infections towards the identification of predictive biomarkers of response, to avoid ineffective therapies and reduce side effects. In the field of infectious diseases, instead, multi-omics studies may lead to the discovery of novel anti-infective treatments and vaccines to prevent the impact of traditional and novel infectious diseases according to population epidemiology, viruses and bacteria genomes, and patients’ pathophysiological hallmarks.

## 5. Conclusions

Diagnosis and treatment of healthcare-associated infections can lead to optimal outcomes for gynecological cancer patients with infections; therefore, implementing this scalable program for a larger number of patients in a broader personalized medicine infrastructure could be strategic to providing integrated complex care management when comorbid conditions are involved.

## Figures and Tables

**Table 1 jpm-12-00650-t001:** Characteristics of study population.

VARIABLES	*N* (%) = 27
Median age	60 (49–70)
Italian nationality	27 (100)
Ward - Gynecologic ward - Gynecologic day hospital - Gynecologic first aid	10 (37.0) 8 (29.6) 9 (33.4)
Oncologic disease - Ovarian cancer - Uterine cancer - Cervical cancer - Other	13 (48.2) 2 (7.4) 8 (29.6) 4 (14.8)
Specific infections - Urinary tract infections - Bloodstream infections - Surgical site infections - Pelvic inflammatory disease - Pneumonia - Rectovesical fistula	12 (44.4) 9 (33.4) 2 (7.4) 2 (7.4) 1 (3.7) 1 (3.7)

## Data Availability

Data supporting our results can be shared upon reasonable request.

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
