# Peer review of "Implementing a Personalized Antimicrobial Stewardship Program for Women with Gynecological Cancers and Healthcare-Associated Infections"

_jpm, 2022, doi:10.3390/jpm12040650_

Round 1

Reviewer 1 Report

An interesting paper asking important questions.  The limitation in terms of number of patients and therefore generalisability is well stated.

Presentation can be improved with a classic 'Table 1' BMJ style giving relevant demographic information such as age, ethnicity, specific cancers, specific infections, other medications, planned or emergency surgery etc.

The participants numbers would be better displayed as a flow chart of all considered and numbers excluded on account of lack of follow-up and other inclusion criteria to give an idea of the underlying population. 

The discussion of big data and AI is anachronistic as you have small data which is not suitable. Suggest remove.

The biggest omission is consideration of the HRT status of these patients.  They are all of menopausal or perimenopausal age and lack of oestrogen is well known to affect succeptibility to UTIs, yet no mention is given of whether these women were taking HRT or had taken it prior to surgery and any advice to discontinue in anticipation of surgery.  Local oestrogen is often prescribed to prevent UTI  see Perratta et al Cochrane systematic review  CD005131.  Please either add HRT status of patients or explain in the limitations why it was excluded & discuss its importance.

Author Response

Dear Reviewer,

thank you for your precious comments that helps improving our manuscript. Please, find our responses in green in the text below.

1) Presentation can be improved with a classic 'Table 1' BMJ style giving relevant demographic information such as age, ethnicity, specific cancers, specific infections, other medications, planned or emergency surgery etc.

We added a table with the clinical information of the study population, as suggested.

2) The participants numbers would be better displayed as a flow chart of all considered and numbers excluded on account of lack of follow-up and other inclusion criteria to give an idea of the underlying population. 

Almost all patients with a health-care associated infection were analyzed. We only excluded patients lost to follow-up (5 patients). Twenty-six patients, evaluated by the infectious disease specialist, were not diagnosed with an infection, so they were not part of the study population. We did not add a flow-chart due to the paucity of data, but we better specified inclusion and exclusion criteria and reported the size of the excluded population in the “Results” section.

3) The discussion of big data and AI is anachronistic as you have small data which is not suitable. Suggest remove.

We have now removed this paragraph, as suggested.

4) The biggest omission is consideration of the HRT status of these patients.  They are all of menopausal or perimenopausal age and lack of oestrogen is well known to affect succeptibility to UTIs, yet no mention is given of whether these women were taking HRT or had taken it prior to surgery and any advice to discontinue in anticipation of surgery.  Local oestrogen is often prescribed to prevent UTI see Perratta et al Cochrane systematic review CD005131.  Please either add HRT status of patients or explain in the limitations why it was excluded & discuss its importance.

We acknowledge your point of view about the importance of HRT as a risk factor for UTIs. In our population, no individual was prescribed with HRT at the moment of infection diagnosis nor prior to surgery. We added this information in table 1.

Reviewer 2 Report

The work is about a brief report regarding implementing a personalized antimicrobial stewardship program for women with gynecological cancers and healthcare-associated infections. The approach of integrating a structural model for each gynecological cancer patient with infections is essential to meet their specific needs, is good so that personalized care can be provided. A personalized care/medicine approach may be practical with small sample size, but it may be challenging to apply it on a large/ very large scale due to issues associated with infrastructure, availability of enough health care professionals, and finally associated costs.

The overall work is good and presented well, but it may take a long time to address the issues associated with this approach.

Author Response

Dear Reviewer,

thank you for your precious comment that helps improving our manuscript.

We acknowledge your kind observation as a really important issue for implementing the described care model. We added the following sentence in the “Discussion” section:

“Moreover, a pharmacoeconomic assessment could help understanding the cost-effectiveness of this program: in fact, costs associated with this personalized approach could become excessive when considering population larger than that included in our study; the availability of dedicated health-care professionals or bigger infrastructures could become other important limits for implementing this care model”.

Thank you

Reviewer 3 Report

The authors report their personal experience in a stewardship program in a specific group of patients such as oncologic patients of gynecologic origin.These programs have proven useful in limiting indiscriminate antibiotic use and improving health outcomes. The authors report the results of the application of this program in terms of clinical cure infection breakthrough or relapse within 30 days, death within 30 days and length of hospital stay.

However, the interest of the present study is limited for several reasons. Among which are: 1) in general, excessive extension in the introduction, discussion or conclusions without expressing the novelty or the importance or the difference of this program in these patients in particular; 2) these programs have already been implemented for some time and have been reported in extensive scientific studies. 3) The objective of the study is to report the experience of the team, but there is no comparison group or a period of time with which to compare and the sample size is excessively limited (the time in which the 27 patients were recruited is unknown),  4) the materials and methods are described in a limited way, for example, the duration of the study or the dates between which it was carried out are not defined, 5) the results are confusing, for example in lines 195-197 "In the remaining 12 cases, the median time of hospital stay was 11 days (IQR 5-12), of which 9 days on antimicrobial therapy (IQR 1-11). However, the median antimicrobial therapy duration was 15 days (IQR 10-17), which indicates a median 5 days-reduction in length of hospital stay (IQR 4-9)."; 6) the discussion does not compare the results obtained by the authors with those of other studies; 7) in the conclusions the authors mix concepts that should be dealt with in the discussion.

Author Response

Dear Reviewer,

thank you for your comments that help improving our manuscript. Please, find our responses in green in the text below.

However, the interest of the present study is limited for several reasons. Among which are:

1) in general, excessive extension in the introduction, discussion or conclusions without expressing the novelty or the importance or the difference of this program in these patients in particular

2) these programs have already been implemented for some time and have been reported in extensive scientific studies.

Thank you for elucidating these important issues. We have discussed the presence of significant literature about these topics, also evidencing the lack of good quality data, especially in the more specific setting of gynecologic cancers. Particularly, we replaced reference 11 with a review of antimicrobial stewardship studies about oncologic patients, that we discussed as follows:

Clinical studies have increased in recent years concerning antimicrobial stewardship programs specifically addressing oncologic populations, which confirm the potential usefulness of several approaches to decrease antibiotic use without increased negative consequences. However, more quality data are needed, especially in the management of more specific cancers, such as gynecologic ones [11].” 

In order to evidence the novelty of our study, we added the following sentence: “To our knowledge, this is the first study exploring the effect of an antimicrobial stewardship program dedicated to gynecological cancer patients. In particular, we highlighted the possibility of implementing more stewardship interventions at the same time, including personalized diagnostic approaches and antibiotic therapy, early antibiotic de-escalation, use of OPAT for reducing hospital stay, and patients’ evaluation by the infectious disease specialist after hospital discharge.”

3) The objective of the study is to report the experience of the team, but there is no comparison group or a period of time with which to compare and the sample size is excessively limited (the time in which the 27 patients were recruited is unknown)

In agreement with your suggestion, we specified the study period in the “Methods” section. In addition, we acknowledged the lack of comparison with the previously practiced approach as a limitation of our study. We added the following sentence in the “Conclusions”:

“First, the small sample size of our population hampers definite conclusions about the effectiveness of this strategy, especially in comparison with previous approaches.”

4) the materials and methods are described in a limited way, for example, the duration of the study or the dates between which it was carried out are not defined

We reported the study period and clarified the exclusion criteria of the study. We also added the sample of the excluded population in the “Results” section.

5) the results are confusing, for example in lines 195-197 "In the remaining 12 cases, the median time of hospital stay was 11 days (IQR 5-12), of which 9 days on antimicrobial therapy (IQR 1-11). However, the median antimicrobial therapy duration was 15 days (IQR 10-17), which indicates a median 5 days-reduction in length of hospital stay (IQR 4-9)"

We better clarified this result by replacing the mentioned sentence with the following one:

“In the remaining 12 cases, the infections required hospitalization or prolongations of the current hospital stay. These infections were represented by BSI in 8 cases (66.7%), UTI and PID in 1 case each (8.3%), SSIs in 2 cases (16.7%). Antimicrobial therapy was administered for a median of 15 days (IQR 10-17), 9 of which (IQR 1-11) during hospital stay. This led to a significant reduction in length of hospital stay (5 days-median reduction, IQR 4-9).”

6) the discussion does not compare the results obtained by the authors with those of other studies

Our work was single-arm, so it was difficult to establish comparison with other stewardship approaches. However, we tried to include other experiences and make some considerations in comparison to our data. Particularly, we added the following sentences in the “Discussion” section:

1) “A previous report by Thursky et al. [12] evidenced the usefulness of implementing an in-tegrated approach for the management of sepsis in a cancer hospital: after the implementation period, the rate of time to antibiotics was halved, the rate of intensive care unit admission was reduced, as well as the overall length-of-stay. Despite a control arm was not present in our study, the possibility of reducing the time of hospital-based antimicrobial administration seems to suggest a similar outcome in gynecologic cancer patients.”.

2) The high percentage of clinical cure and lack of infection relapse reported in our study seems to be in line with other experiences: despite differences in antimicrobial stewardship approaches, the rate of mortality or unfavorable outcomes (for instance ICU admission, re-escalation of antibiotics) were not higher in stewardship programs compared with control groups, according to a recent review of literature [11].  

7) in the conclusions the authors mix concepts that should be dealt with in the discussion

We agree with your position. We redefined the Discussion by including the significant points of the Conclusions. We limited the latter by only including the final statement that summarizes the importance of the antimicrobial stewardship program in the setting of gynecologic cancer patients.